# The phase separation underlying the pyrenoid-based microalgal Rubisco supercharger

Tobias Wunder[1], Steven Le Hung Cheng[1], Soak-Kuan Lai[1], Hoi-Yeung Li[1] & Oliver Mueller-Cajar[1]

The slow and promiscuous properties of the $CO_2$-fixing enzyme Rubisco constrain photosynthetic efficiency and have prompted the evolution of powerful $CO_2$ concentrating mechanisms (CCMs). In eukaryotic microalgae a key strategy involves sequestration of the enzyme in the pyrenoid, a liquid non-membranous compartment of the chloroplast stroma. Here we show using pure components that two proteins, Rubisco and the linker protein Essential Pyrenoid Component 1 (EPYC1), are both necessary and sufficient to phase separate and form liquid droplets. The phase-separated Rubisco is functional. Droplet composition is dynamic and components rapidly exchange with the bulk solution. Heterologous and chimeric Rubiscos exhibit variability in their tendency to demix with EPYC1. The ability to dissect aspects of pyrenoid biochemistry in vitro will permit us to inform and guide synthetic biology ambitions aiming to engineer microalgal CCMs into crop plants.

[1] School of Biological Sciences, Nanyang Technological University, 60 Nanyang Drive, Singapore 637551, Singapore. Correspondence and requests for materials should be addressed to O.M.-C. (email: cajar@ntu.edu.sg)

The efficiency of photosynthesis is frequently limited by the slow and error-prone kinetic properties of Rubisco[1]. This necessitates extreme overexpression of the protein in plant leaves to overcome the former, and substantial loss of energy when recycling the oxygenation product 2-phosphoglycolate[2]. In photosynthetic micro-organisms these shortcomings are compensated by powerful $CO_2$-concentrating mechanisms (CCMs) that sequester Rubisco in microcompartments, allowing its active sites to be saturated by $CO_2$ that is delivered there by active transport[3,4]. A major objective in Synthetic Biology efforts to enhance the photosynthetic efficiency of crop plants concerns the introduction of the microalgal or cyanobacterial CCMs into higher plant chloroplasts[5–7]. The heartpiece of the CCM of most eukaryotic microalgae consists of an electron-dense Rubisco-containing compartment known as the pyrenoid[8].

Substantial recent advances have revealed the pyrenoid of the model alga *Chlamydomonas reinhardtii* to behave as a phase-separated liquid compartment[9] similar to other non-membraneous organelles, such as P-granules, nucleoli, and Cajal bodies[10–12]. In addition to Rubisco, the pyrenoid contains an abundant repeat-protein EPYC1 (also known as LCI5[13]), which is thought to act as a scaffold for other pyrenoid components[14]. Importantly, the *epyc1* mutant, which contains severely reduced levels of this protein, requires high $CO_2$ for photoautotrophic growth and is unable to effectively concentrate $CO_2$. The pyrenoid of *epyc1* is of decreased size, matrix density is reduced, and the majority of Rubisco is localized to the chloroplast stroma[14]. Two recent studies report that pyrenoid composition is highly complex, with 89 and 190 proteins respectively being proposed to occupy this compartment[15,16].

Here we use biochemical reconstitution to demonstrate that Rubisco and EPYC1 are the two components necessary and sufficient to bring about a liquid-liquid phase separation (LLPS) that recapitulates the liquid-like behavior reported for the microalgal pyrenoid[9]. The data generated using this minimal system provide critical information to guide the engineering of algal CCMs into plants to achieve higher photosynthetic efficiency[17].

## Results

**EPYC1 and Rubisco are necessary and sufficient for LLPS.** The behavior of pyrenoid localized fluorescent fusion proteins in the *C. reinhardtii* chloroplast in conjunction with Rubisco distributions determined by in situ cryo-electron tomography[9,14,18] have culminated in a current LLPS model of pyrenoid formation. Briefly, the EPYC1 protein consists of four quasi-identical 60 amino-acid tandem repeats, which are predicted to bind multivalently to the surface of Rubisco holoenzymes (of $L_8S_8$ stoichiometry) and consequently drive a phase separation analogous to that described for other systems[19,20].

To enable mechanistic experiments to dissect this phenomenon we produced in *Escherichia coli* and purified to homogeneity EPYC1 (residues 46–317, excluding a predicted transit peptide[21]) a suite of EPYC1 segments and green fluorescent protein (GFP) fusion proteins. A range of phylogenetically diverse Rubisco proteins were also produced recombinantly or purified from source (Supplementary Fig. 1a). To overcome extensive proteolytic degradation of EPYC1 proteins that was initially encountered, we eventually produced them as His$_6$-Ubiquitin fusion protein[22] while overexpressing the endogenous chaperonin GroEL/ES (Supplementary Fig. 1b). In high-salt buffer (500 mM NaCl) the protein eluted as a concentration-independent species with a Stokes radius of 3.1 nm from a sizing column. At 250 mM NaCl EPYC1 appeared to interact with the resin (Supplementary Fig. 1c). Consistent with predictions from primary sequence

analysis[14], the circular dichroism spectrum indicates EPYC1 to be intrinsically disordered (Supplementary Fig. 1d)[23].

Mixing pure *C. reinhardtii* Rubisco (CrRubisco) and EPYC1 led to immediate formation of a turbid solution that cleared over time (Fig. 1a). The turbidity was caused by the formation of spherical droplets from the bulk solution that could be labeled by including a fluorescent EPYC1-GFP fusion protein in the reaction (Fig. 1b) and occurred within seconds (Supplementary Fig. 2a). The observed clearance of the solution was caused by fusion of the droplets into a large homogeneous droplet (coalescence), supporting their liquid nature[12] (Fig. 1c, Supplementary Movie 1). Demixed droplets could be harvested by centrifugation, and SDS-polyacrylamide gel electrophoresis (SDS-PAGE) analysis confirmed that both EPYC1 and Rubisco had entered the droplets (Fig. 1d). EPYC1-GFP presented with similar properties regarding Rubisco demixing as the unfused protein (Supplementary Fig. 2b, c). To more closely resemble the native system, we subsequently employed reactions doped with trace amounts of fluorescent protein (1–5%). Droplets could also be labeled by including the closely related (81% large subunit sequence identity) cyanobacterial Rubisco-GFP (SeRubisco-GFP) in the reaction (Supplementary Fig. 2d, e). As observed in multiple other biological systems[24,25], phase separation was protein concentration-dependent and salt (but not detergent)-sensitive (Fig. 1b, d, e, Supplementary Fig. 2f, g). Hence ionic interactions are important in the Rubisco-EPYC1 association. In contrast to other phase-separating scaffold proteins that contain unstructured regions[26–29], demixing of purified EPYC1 or Rubisco alone did not occur even at low temperatures and high concentrations (Supplementary Fig. 2h, i). Only at 100–200 μM EPYC1 (2.8–5.5 mg/ml) could small droplets be seen when the molecular crowding agent polyethyleneglycol-3350 (10% w/v) was present. At 200 μM ~10% of the EPYC1 protein pelleted in the sedimentation assay (Supplementary Fig. 2h, i).

To observe efficient LLPS the presence of both Rubisco and EPYC1 was necessary, consistent with complex (rather than simple) coacervation[30,31]. Rubisco-EPYC1 demixing was not significantly affected by temperature within the tested range of 0–40 °C (Supplementary Fig. 2j). Consistent with its disordered nature, EPYC1 could be boiled for 15 min, while remaining soluble and retaining the ability to subsequently phase-separate with Rubisco (Supplementary Fig. 2k).

**Phase-separated Rubisco is functional.** To become catalytically functional, Rubisco active sites need to bind $CO_2$ and $Mg^{2+}$ cofactors to form the holoenzyme termed ECM. To confirm that Rubisco structure and function was not negatively affected by concentrating the protein in the demixed droplets we assayed the ECM holoenzyme prior to, and after phase separation. Resuspending the sedimented droplets in buffer containing 300 mM NaCl resulted in recovery of >85% of the functional holoenzyme (Fig. 2a). We then chose to compare Rubisco activity in solution to the demixed form using a two-step discontinuous assay. ECM has an extraordinarily high affinity for the carboxylation transition state analog carboxyarabinitol bisphosphate[32] (CABP), permitting dead-end inhibited complexes (ECMC) to be formed. We incubated ECM in the presence and absence of EPYC1, added its substrate RuBP, and stopped the reaction at various time points using an excess of CABP. The carboxylation product 3PG was then quantified. Soluble and phase-separated Rubisco produced 3PG at the same rate, showing that the demixed enzyme is catalytically active (Fig. 2b). Microscopy and the droplet sedimentation assays confirmed that the Rubisco was droplet-bound during this experiment (Supplementary Fig. 3a, b).

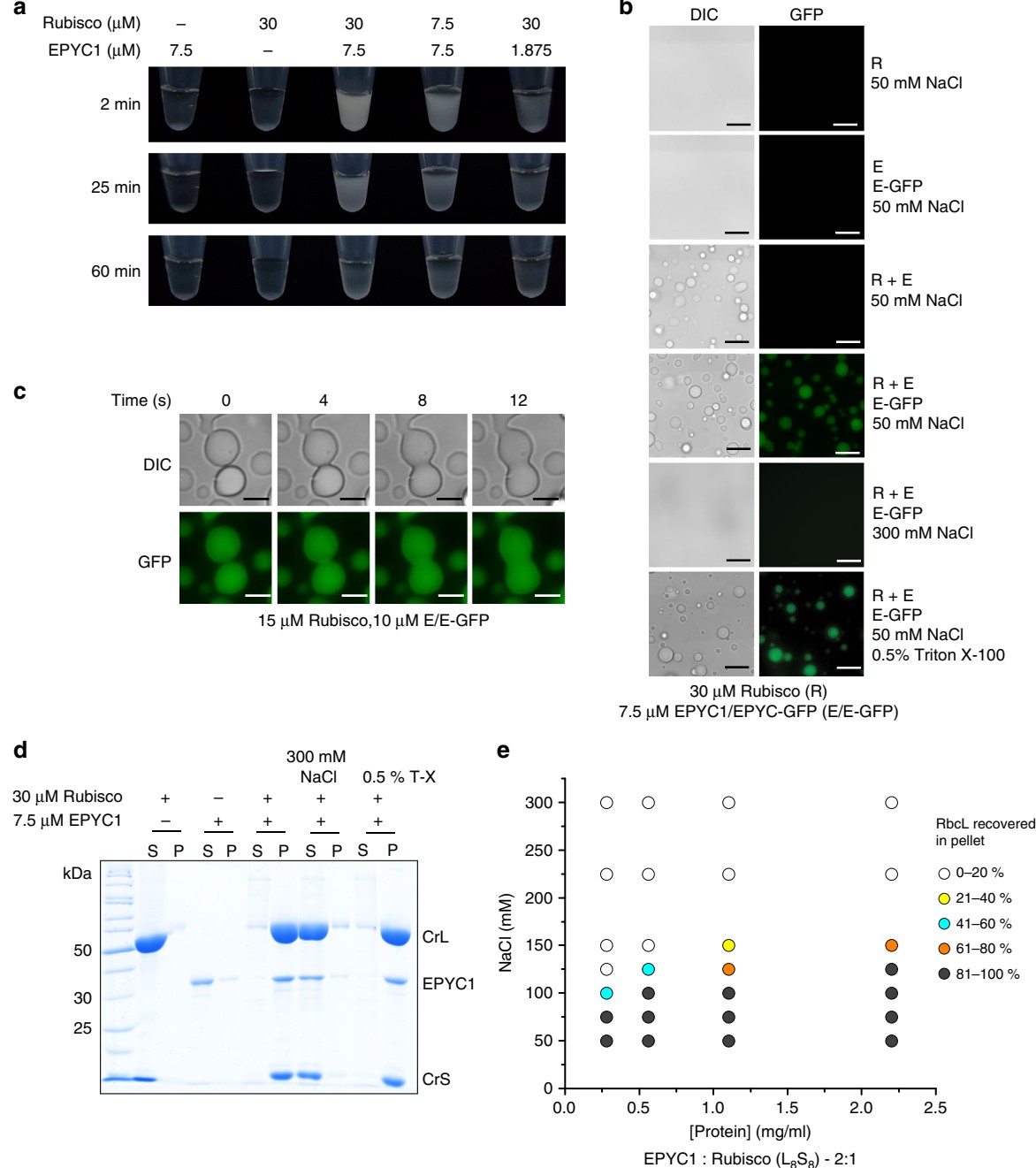

**Fig. 1** Rubisco and EPYC1 are necessary and sufficient to bring about a liquid-liquid phase separation. **a** Addition of EPYC1 to Rubisco results in turbidity that clears over time. **b** The turbidity is caused by the formation of spherical droplets that are fluorescently labeled when 5% EPYC1-GFP is included. Scale bar 15 μm. **c** Droplets fuse by coalescence. Scale bar 5 μm. **d** The droplets can be harvested by centrifugation (10 min, 21,100 × g). **e** Protein demixing is protein and salt concentration-dependent. The phase diagram was generated by quantifying the proportion of RbcL found in the pellet fraction. All experiments were carried out in 20 mM Tris-HCl (pH 8.0) and 50 mM NaCl (buffer A) unless indicated otherwise

**Droplet content is dynamic and components exchange rapidly.** To determine the range of permissible stoichiometries in droplets, we titrated both components followed by densitometric quantification of the resultant pellet fraction composition (Supplementary Fig. 4a). We found 7.5 μM EPYC1 was able to completely demix 30 μM Rubisco active sites (Fig. 3a), corresponding to a ratio of two EPYC1 molecules per Rubisco holoenzyme. However, as expected for phase-separated systems droplet composition was not fixed[20], and the ratio of EPYC1:Rubisco found in the droplets varied from 1 to 4 (Fig. 3b).

Fluorescence recovery after photobleaching (FRAP) experiments were used to measure the mobility of droplet components. To capture the relative dynamics of Rubisco and EPYC1 we formed Rubisco-EPYC1 droplets labeled with either EPYC1-GFP or SeRubisco-GFP (comprising <1.5% of either EPYC1 or Rubisco) to achieve similar epifluorescence intensities. SeRubisco-GFP droplets achieved 50% of the original fluorescence intensity after 40 s (Fig. 3c). In contrast EPYC1-GFP in droplets containing 1% (or even 100%) of the fusion protein displayed a recovery half-time of ~10 s (Fig. 3c, Supplementary

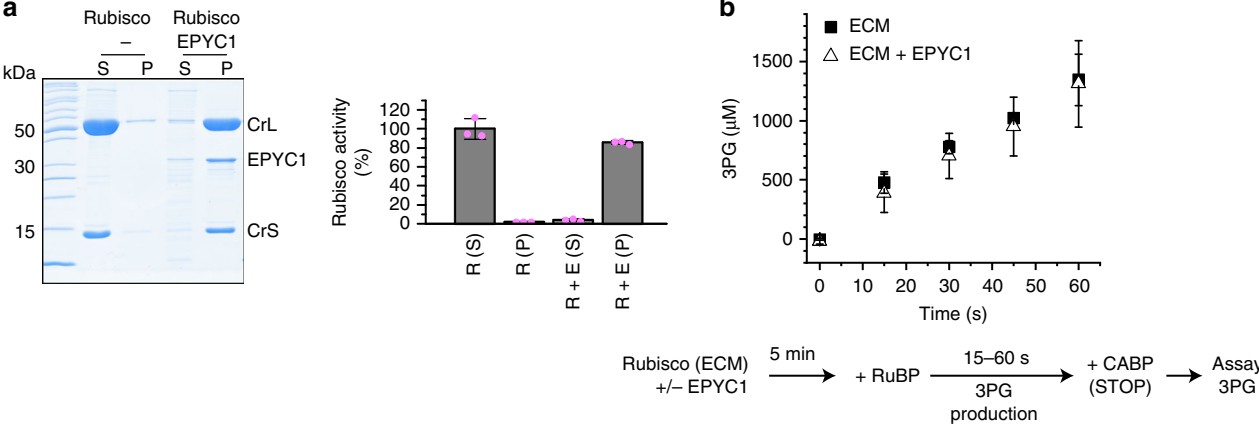

**Fig. 2** Demixed Rubisco remains active. **a** Rubisco remains functional following droplet formation. Pelleted droplets were resuspended in one volume of ECM buffer containing 300 mM NaCl and assayed for Rubisco activity. **b** Rubisco enzyme activity (indicated as 3PG production) is unaffected by droplet formation. Carboxylation reactions contained 7.5 μM Rubisco in presence and absence of EPYC1 (3.75 μM). Error bars indicate s.d. of three technical replicates

Fig. 4b). When entire droplets were photobleached, similar recovery kinetics were observed, indicating rapid interchange of the droplet components with the bulk phase (Supplementary Fig. 4c). Consistently, addition of the GFP fusion proteins 3 min after droplet formation also resulted in rapid droplet labeling (Supplementary Fig. 4d).

To further study the compositional interchange of the droplets we assessed the speed at which the inhibited ECMC exchanged with the functional ECM holoenzyme. ECM was used to form the droplets, which were perturbed with an equimolar amount of the inhibited enzyme after 3 min (Fig. 3d). The subsequent release of ECM from the droplets was then quantified by assaying relative Rubisco activity in the supernatant following droplet sedimentation (Fig. 3e). The two pools of active and inhibited Rubisco equilibrated within 10 min with a half-time of ~100 s (Fig. 3f). In contrast, exchange of EPYC1 with FLAG-EPYC1, as assessed by similar experiments, was too rapid to resolve by the droplet sedimentation assay (Supplementary Fig. 4e). Overall, the mobilities of droplet components reported here are qualitatively consistent with the recovery kinetics reported for pyrenoid-localized Venus-fused EPYC1 and Rubisco in vivo[9].

**Different Rubiscos vary in their ability to phase-separate**. To understand the determinants of the pyrenoid LLPS we tested a range of diverse prokaryotic (produced recombinantly in *E. coli*) and eukaryotic Rubisco enzymes (purified from source) for their ability to demix with EPYC1 by droplet sedimentation and microscopy (Fig. 4a, b, Supplementary Fig. 5 a, b). Cyanobacterial SeRubisco (*Synechococcus* PCC6301) formed droplets that most closely resembled the endogenous combination. The proteobacterial carboxysome-associated Af2Rubisco (*Acidithiobacillus ferrooxidans*, carboxysomal gene cluster-encoded Form I Rubisco) required high EPYC1 concentrations to phase-separate, whereas the cytosolic Form I Rubisco encoded by the same organism (Af1—equivalent to AfLS[33]) failed to demix. Af1Rubisco purified in the absence of its small subunits (L8) phase separated, but formed needles instead of droplets. Other enzymes, such as higher plant rice (Os—*Oryza sativa*), more remote Form II (AfM—*A. ferrooxidans* Form II Rubisco[33]), or red-type Form I Rubiscos (Rs—*Rhodobacter sphaeroides* 2.4.1[34]) failed to demix with EPYC1 (Fig. 4a, b, Supplementary Fig. 5 a, b).

It has been shown that *Chlamydomonas* large subunits assembled with small subunits from plants, or mutated at surface exposed α-helices, were unable to form the pyrenoid in vivo[35].

potentially implicating these elements in the EPYC1-Rubisco interaction. However, EPYC1 and purified *Chlamydomonas* Rubisco small subunits (CrRbcS) did not phase-separate (Supplementary Fig. 5c, d), possibly because its monomeric nature precluded multivalent interactions. We therefore utilized the cyanobacterial RbcL8 core (SeL8—lacking small subunits) from *Synechococcus* PCC6301 as a scaffold for the small subunits of *Chlamydomonas* or rice. Consistently, chimeric SeL8CrS8 demixed well with EPYC1, whereas SeL8OsS8 required high EPYC1 concentrations to do so (Fig. 4c, d, Supplementary Fig. 5e,f). The droplets formed with chimeric SeL8OsS8 were smaller and did not fuse productively (Fig. 4d).

The SeL8 core alone demixed well with EPYC1, but formed abnormal structures resembling tiny clumped droplets. The SeL8-EPYC1 droplets were similarly salt sensitive as the wild-type system (Supplementary Fig. 5g, h). Addition of purified algal small subunits (CrRbcS) to such droplets resulted in a rapid restoration of normal morphology (Fig. 4e, Supplementary Fig. 6a–c, Supplementary Movie 2), showing that small subunits entered the droplets and SeL8 cores were in a conformation conducive to accept them. FRAP assays of droplets generated with the chimeric enzymes or the SeL8 core indicated similar or increased EPYC1-GFP mobility compared with the wild-type system (Supplementary Fig. 6d, e). The observed demixing of the SeL8 core could be driven either by specific interactions between EPYC1 and the Rubisco large subunit or nonspecific interactions with the surface covered by the small subunit in the holoenzyme. Adding an EPYC1 fragment (EPYC1Δ2-4, containing the first repeat and the C terminus, Supplementary Table 1) to Rubisco followed by native-PAGE results in a smear and reduced mobility for CrLS, but not for OsLS, and thus provides a protein-binding assay (Fig. 4f, Supplementary Fig. 7). Note that EPYC1Δ2-4 alone does not enter the gel due to its positive charge. This gel-shift assay indicated that both AfL8 and SeL8 cores interacted more strongly with EPYC1Δ2-4 than the respective L8S8 holoenzymes (Fig. 4f, Supplementary Fig. 7). This comparison implies that the newly exposed large subunit surface is likely to contribute to the interaction in both cases.

**The importance of EPYC1 multivalency**. Similar to other LLPS systems[19,20,36], the EPYC1 scaffold contains several repeating units (Supplementary Fig. 8a), which Freeman Rosenzweig et al.[9] predicted to bind the oligomeric Rubisco via multiple low-affinity interactions to drive droplet formation and permit internal

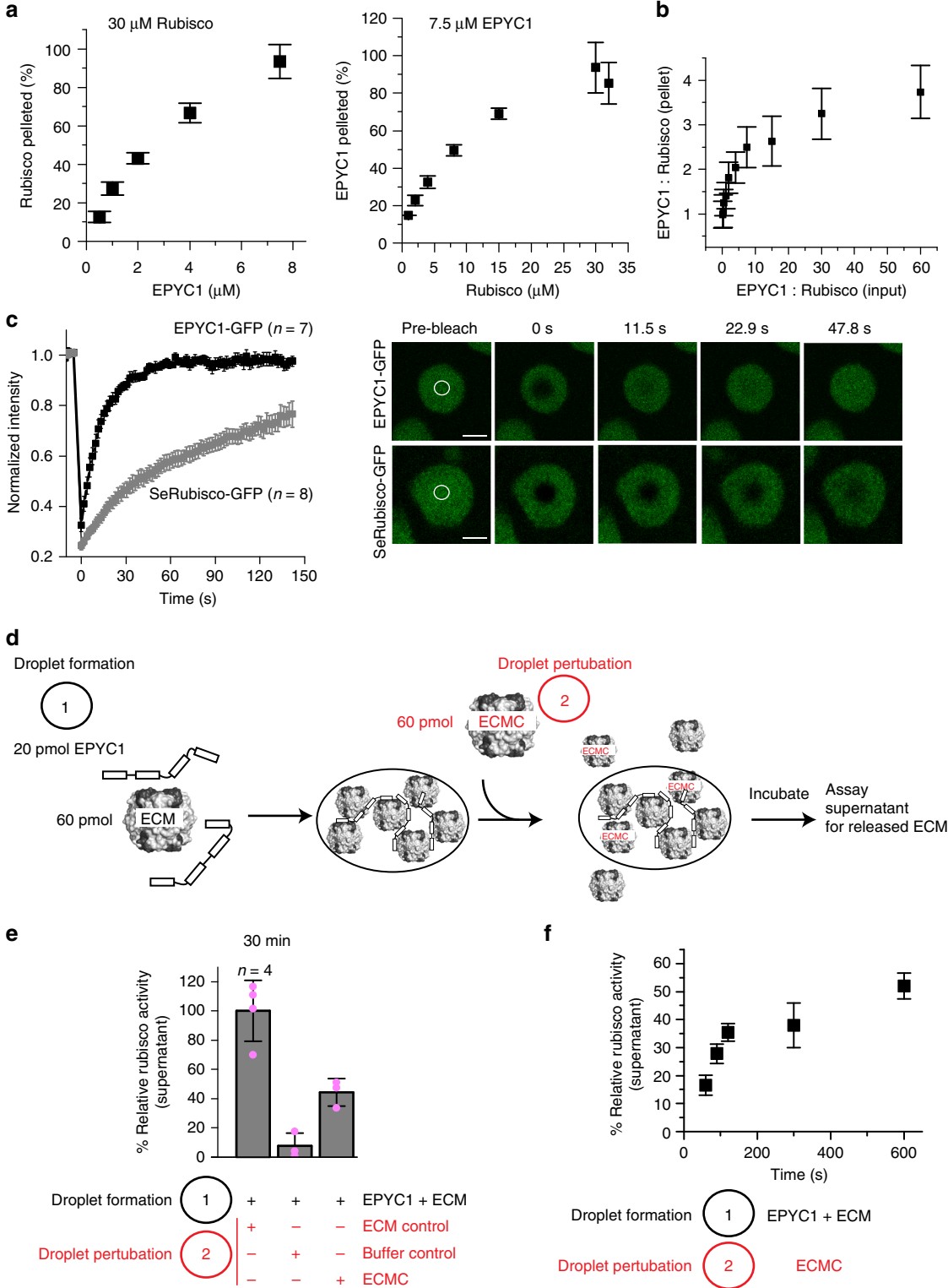

**Fig. 3** Droplet composition is dynamic. **a** EPYC1 (left) and Rubisco (right) were titrated followed by SDS-PAGE and densitometric analysis of droplet composition. 30 μM Rubisco:7.5 μM EPYC1 (1 $L_8S_8$:2 linker proteins) resulted in complete demixing. **b** EPYC1:Rubisco stoichiometries in droplets range from 1 to 4. **c** Fluorescence recovery after photobleaching of demixed droplets (15 μM Rubisco and 10 μM EPYC1) labeled with either 0.1 μM EPYC1-GFP or 0.2 μM cyanobacterial Rubisco-GFP (SeRubisco-GFP). Error bars indicate the s.e.m. Scale bar 2 μm. **d** Biochemical Rubisco exchange assay scheme. ECM active Rubisco, ECMC dead-end inhibited Rubisco. Droplets are formed using ECM (indicated as droplet formation 1), and an equal amount of ECMC is added after 3 min (indicated as droplet perturbation 2). Displaced Rubisco is measured by Rubisco assay of the supernatant. **e** Addition of non-functional ECMC to droplets formed with functional ECM results in displacement of ECM from the droplets. **f** ECM displacement kinetics. Error bars excluding **c** indicate mean and s.d. ($n = 3$ unless indicated differently)

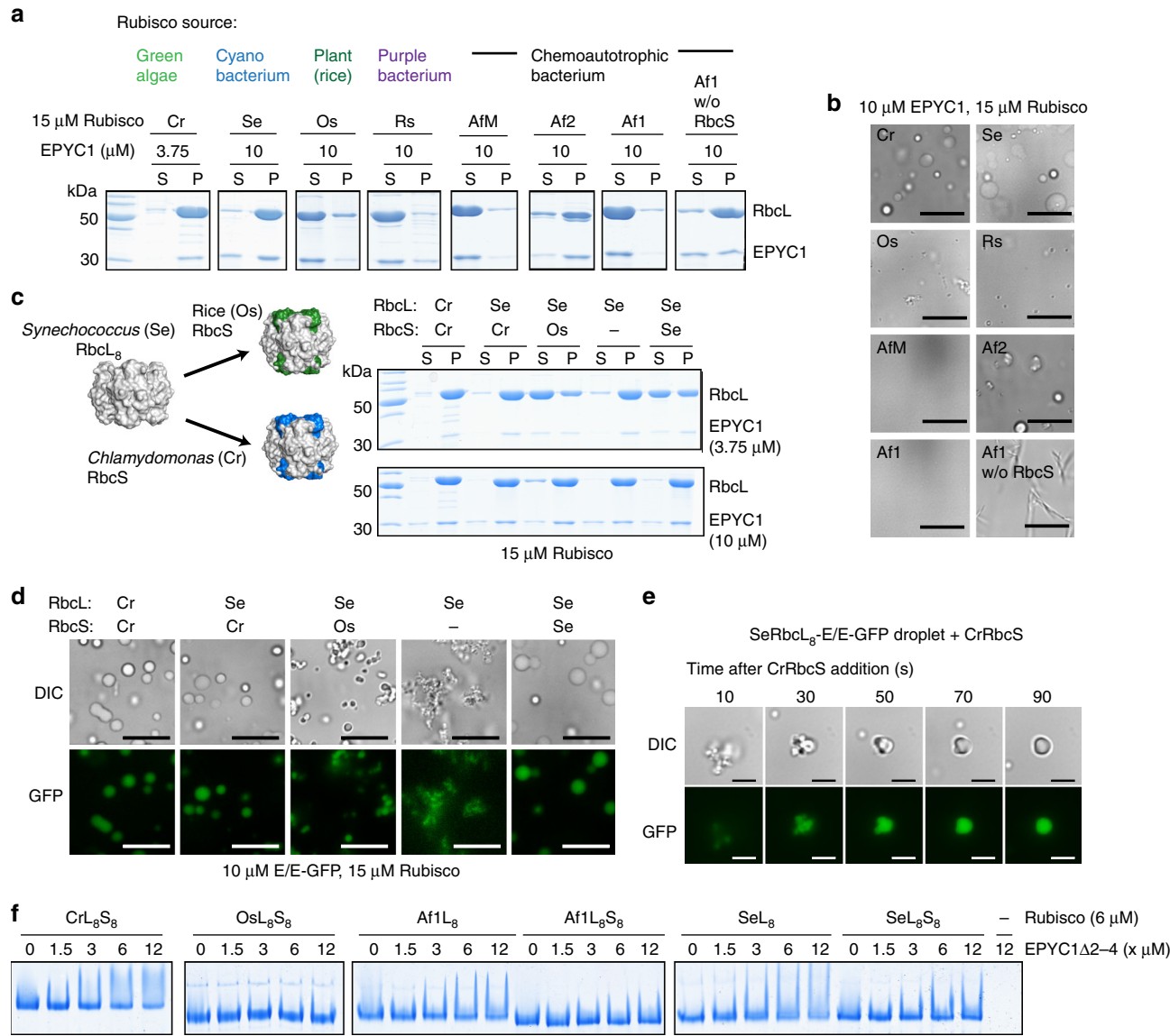

**Fig. 4** Diverse Rubisco enzymes vary in their tendency to demix with EPYC1. **a, b** The tendency of a phylogenetically diverse set of Rubisco enzymes to demix was assessed using the sedimentation assay (**a**) and microscopy (**b**). Scale bar 15 μm. **c, d** The role of the small subunit in demixing was assessed using chimeric Rubiscos (*Synechococcus* large subunit cores (L$_8$) either lacking, or assembled with small subunits from rice (Os) or *Chlamydomonas* (Cr)). Sedimentation (**c**) and microscopy (**d**) assays are shown. Scale bar 15 μm. **e** Addition of external CrRbcS (4 μM) rapidly changes the appearance of SeL$_8$ (2.5 μM)-EPYC1/EPYC1-GFP (1.25/0.1 μM) droplets. Scale bar 5 μm. **f** Native PAGE gel-shift assay using Rubisco and the single-repeat variant EPYC1Δ2-4 indicates increased binding for Rubiscos lacking the small subunit

rearrangements consistent with their liquid property. The same work proposes on theoretical grounds that four EPYC1 repeats (and Rubisco-binding sites) would correspond to a magic number that would lead to smaller aggregates, than either three or five repeats. To test the importance of EPYC1 multivalency for phase transition, we therefore produced a series of EPYC1 derivates containing one to five repeats (Fig. 5a). Somewhat unexpectedly, even the single-repeat EPYC1Δ2-4 was able to demix with Rubisco (Fig. 5b, Supplementary Fig. 8b), suggesting this minimal fragment still contributes at least two Rubisco-binding sites. However, droplets formed using lower-valency fragments were flatter upon contacting the coverslip than those containing more repeats (Fig. 5c, Supplementary Fig. 8c, d) indicating that the valency affects droplet surface tension[26]. Additionally, EPYC1-GFP spiked into droplets formed with EPYC1Δ2-4 displayed higher mobility as measured by FRAP

(Fig. 5d). EPYC1 variants containing three to five repeats were able to sediment Rubisco equally well as EPYC1 in our system and we could thus not detect the hypothesized magic number effect[9]. Even the variant containing two repeats (EPYC1Δ2/3) allows almost EPYC1-like demixing, when corrected for the number of EPYC1 repeats present. In contrast the minimal construct EPYC1Δ2-4 required high Rubisco concentrations to effectively demix the enzyme (Fig. 5e, Supplementary Fig. 8e, f). In summary, higher numbers of interaction sites in longer EPYC1 variants shift the phase diagram, permitting demixing to occur at lower protein concentrations.

**Discussion**

In this contribution we provide evidence that Rubisco rapidly forms dense liquid droplets when exposed to the pyrenoidal Rubisco linker protein EPYC1. The fact that the droplets are not

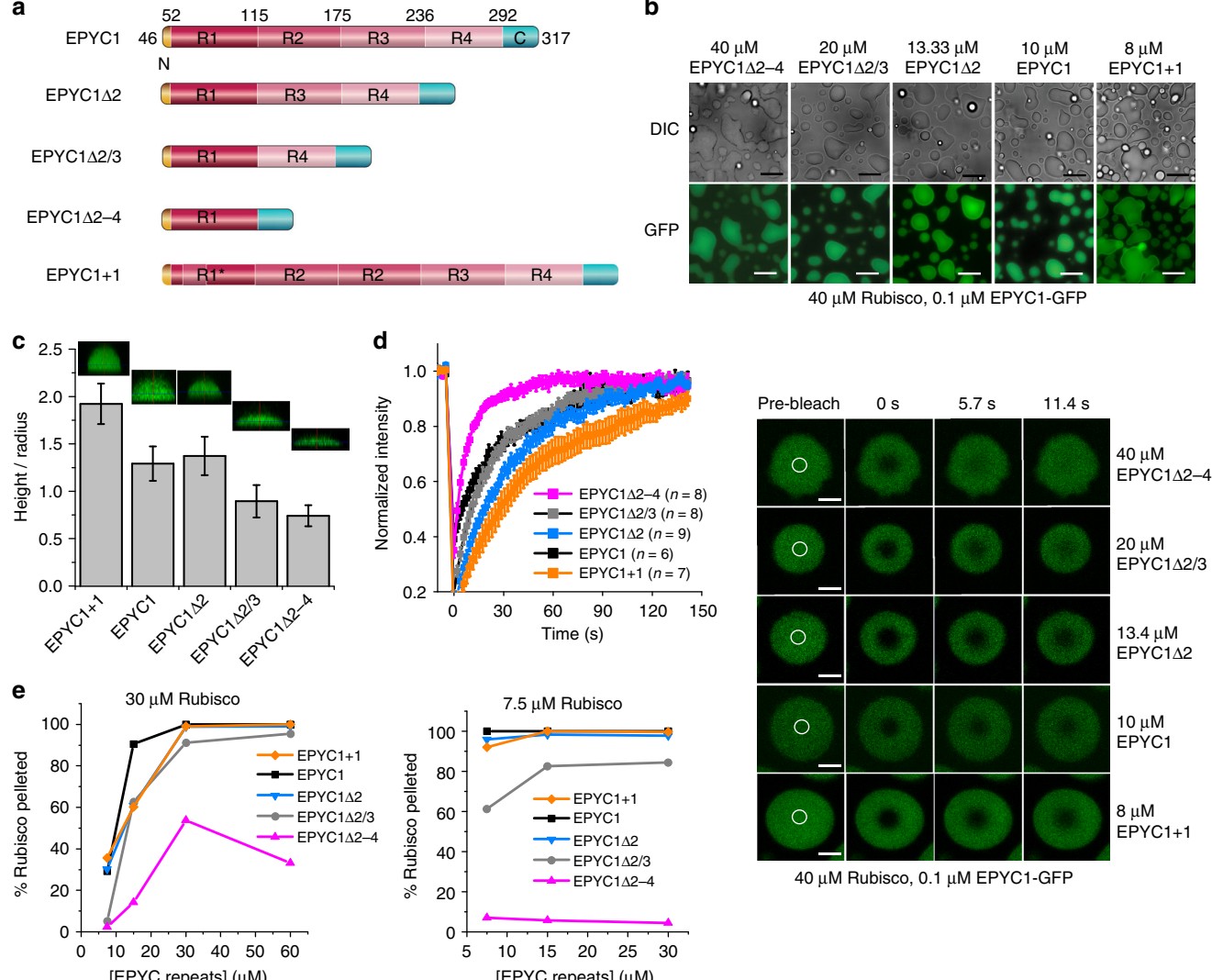

**Fig. 5** The role of the EPYC1 tandem repeats. **a** Schematic representation of EPYC1 constructs including deletions or additions of the repeat regions. Note, R1* indicates a chimeric repeat containing a short R2 sequence insertion. **b** All tested constructs are able to demix Rubisco. Three-minute incubation time prior to imaging within the subsequent 2–6 min. Scale bar 15 μm. **c** EPYC1 proteins with fewer repeats form flatter droplets than variants with higher valency; conditions as in **b**. Error bars indicate the s.d. ($n$ = 42–50; data points shown in Supplementary Fig. 8d). **d** Droplets containing the lowest-valency construct recover more rapidly from photobleaching. Sample size $n$ and s.e.m. indicated. Scale bar 2 μm. **e** Lower-valency (one to two repeats) EPYC1 variants require higher Rubisco concentrations to phase-separate as measured using the droplet sedimentation assay. EPYC1 variants containing three, four, or five repeats behave similarly

limited in size and far larger than the pyrenoid (~1–2 μm diameter) suggest that in vivo the size of the compartment is likely component limited. EPYC1 was predicted to either assemble into a pyrenoid scaffold to which Rubisco binds, or form a co-dependent network with the enzyme[14]. In contrast to the scaffold hypothesis we find EPYC1 is relatively soluble and only homotypically phase separates at concentrations of 100 μM or more in the presence of a crowding agent. To achieve efficient phase separation at lower concentrations both EPYC1 and Rubisco are essential, supporting the co-dependent network model. The two proteins are oppositely charged and phase separation is salt-sensitive. Taken together these observations are consistent with the phenomenon of complex coacervation, where positively and negatively charged polymers interact via electrostatic interactions to form a dense droplet phase of complexed polyions[31]. Indeed, a comparison of the surface charge of the available Rubisco crystal structures indicates the presence of large negatively

charged patches that could mediate such interactions (Supplementary Fig. 9a). One such surface is found at the interface between two large subunit dimers and is particularly pronounced for the cyanobacterial enzyme. When comparing OsRubisco and CrRubisco, the most striking difference is found regarding the surface charge of the small subunit, with the algal Rubisco having a negatively charged top, whereas the plant Rubisco top is neutral or positively charged (Supplementary Fig. 9a). Accordingly, our data indicate that the nature of the Rubisco small subunit can significantly influence and interfere with droplet formation as observed previously in vivo[35].

An analysis of EPYC1 charge distribution using the CIDER[37] webtool highlights positively charged regions that may be involved in interactions with the negative patches on Rubisco, in particular a SKKAV motif located at the end of repeats one, two, and three, as well as the C-terminal region[14] (Supplementary Fig. 9b).

The demixed Rubisco was found to be as active as the soluble enzyme (Fig. 2). Importantly, the physiological CCM function does not require enhanced Rubisco kinetics in the pyrenoid, as it relies on increased $CO_2$ substrate supply. This is achieved by a combination of active transport of inorganic carbon and strategic positioning of carbonic anhydrases[38].

Our first steps toward dissecting the relative importance of EPYC1's 60 amino-acid tandem repeats indicate that higher valency linker proteins enable phase separation to occur at lower protein concentrations. However, the phase-separating behavior of the single-repeat fragment EPYC1Δ2-4 strongly suggests that EPYC1 contains more than the hypothesized four Rubisco-binding sites that have been proposed to be significant on theoretical grounds regarding a magic number effect[9]. Reducing the number of repeats to three did not increase the propensity to phase-separate, which also contradicts this theory. Permutations of the minimal fragment EPYC1Δ2-4 will allow further dissection of the pyrenoid LLPS.

Our reconstituted system is highly relevant to efforts aiming to introduce the algal CCM into higher plants[17]. Our data imply that suboptimal EPYC1-Rubisco interactions will lead to aberrant phase-separating properties that could bring about scenarios such as irreversible aggregation similar to that seen in disease-associated variants of the ALS protein FUS[27]. Formation of droplets with non-native properties in vivo will likely interfere with CCM functioning. Using our system we can rapidly evaluate the phase-separating characteristics of proposed combinations. Finally, we anticipate that combining our minimal, bottom-up reconstitution approach with the large-scale protein-protein interaction roadmap generated by Jonikas and colleagues[15] and the recent successes in producing eukaryotic Rubiscos in bacteria[39] will enable rapid elucidation and dissection of pyrenoid structure and function.

## Methods

**Plasmids.** Supplementary Table 1 shows the plasmids used in this study. All *C. reinhardtii* genes were amplified from cDNA isolated from strain CC-2677 cw15 nit1-315 mt-5D (Chlamydomonas Resource Center, University of Minnesota) and cloned between the *Bam*HI-*Hin*dIII sites of pHue[22] to give pHue*CrEPYC1* or the *Nde*I-*Hin*dIII of sites pET22b (Novagen) to yield pET22b*CrRbcS*. Plasmids encoding fragments and fusion proteins of EPYC1 were generated from *epyc1* templates (subcloned or cDNA) using a combination of site-directed mutagenesis and restriction-free cloning[40] to encode the sequences shown in Supplementary Table 2. pTrc*SynLF345I* was constructed by cloning the *SynLF345I* (synthesized by GenScript) between *Nco*I and *Hin*dIII of pTrcHisB. pTrc*SynLF345IS-GFP* was constructed from pTrc*SynLF345IS* to encode SeRbcS-GFP in place of SeRbcS (Supplementary Table 2). *OsRbcS* was synthesized by GenScript and cloned between *Nde*I-*Hin*dIII sites of pET22b to give pET22b*OsRbcS*. *AfcbbL* was amplified from pET30b*AfcbbLS* and cloned between the *Bam*HI and *Hin*dIII sites of pHue (resulting in the presence of one additional N-terminal serine following His6-Ub cleavage[22]). The genes encoding carboxysomal *A. ferrooxidans* Form I Rubisco (Af2LS) were amplified together (including intergenic region) from genomic DNA[33] and cloned between *Nde*I and *Hin*dIII sites of pET24b to give pET24b*Af2cbbLS*. All protein encoding sequences were verified by DNA sequencing.

**Proteins.** Concentrations of pure protein samples in this study were calculated from absorbance measurements at 280 nm, using extinction coefficients obtained using the ExPASy-ProtParam tool[41]. Prior to experiments, proteins were buffer-exchanged into buffer A (20 mM Tris-HCl, pH 8.0, 50 mM NaCl) using Micro Bio-Spin 6 (BioRad) chromatography columns

EPYC1 protein was produced in *E. coli* strain BL21 (DE3) cells harboring pHue*CrEPYC1* and the *E. coli* chaperonin-encoding pBADESL[42]. Cells were grown in Lysogeny broth medium to an OD600 of 0.8 at 37 °C before initiating the overexpression of the chaperonin by adding 0.2% (w/v) L-arabinose. After 20 min, induction of EPYC1 protein production with 1 mM isopropyl β-D-1-thiogalactopyranoside for 3 h followed. Cells were lysed in high-salt lysis buffer (20 mM Tris-HCl, pH 8.0, 500 mM NaCl, and 10 mM imidazole) in the presence of 2 mM phenylmethanesulfonyl fluoride (PMSF) by ultrasonication following a 30 min lysozyme (0.3 mg/ml) pretreatment. The soluble fraction was loaded on an IMAC (HisTrap™ HP 5 ml, GE Healthcare) column, was washed with high-salt wash buffer (20 mM Tris-HCl, pH 8.0, 500 mM NaCl, and 25 mM imidazole),

and bound proteins were eluted using an imidazole gradient. The N-terminal His6-ubiquitin moiety was cleaved[43] and the protein solution dialyzed against 20 mM Tris-HCl (pH 8.0) and 500 mM NaCl before exposing it again to IMAC, this time collecting the untagged flowthrough. Finally, EPYC1 was subjected to size-exclusion chromatography (Superdex 200 16/600, GE Healthcare) using 20 mM Tris-HCl (pH 8.0) and 500 mM NaCl as the eluent. Glycerol was added to 5% v/v prior to flash freezing and storage at −80 °C. EPYC1 fragments and fusion proteins were also produced in *E. coli* harboring the relevant plasmids (Supplementary Table 1) and purified using the same protocol with the following abbreviations. FLAG-EPYC1 and EPYC1Δ2/3 were not exposed to the second IMAC step. The size-exclusion step was omitted for EPYC1-GFP. For EPYC1Δ2-4 the SEC and IMAC steps were reversed. Buffers used to purify EPYC1-GFP and EPYC1Δ2-4 all contained 250 mM NaCl.

Rubisco was purified from *C. reinhardtii* cells (CC-2677 cw15 nit1-305 mt-5D, Chlamydomonas Resource Center) grown in Sueoka's high-salt medium[44] under airlift conditions. Cells were disrupted by ultrasonication in lysis buffer (100 mM Tris-HCl, pH 8.0, 100 mM NaCl, 0.5 mM EDTA, pH 8.0, and 5 mM dithiothreitol (DTT)) supplemented with 2 mM PMSF and Pierce™ Protease Inhibitor Tablets, EDTA-free (Thermo Scientific). The soluble lysate was fractionated by anionexchange chromatography (ReSource 30Q, GE Healthcare) using a linear salt gradient from 50 mM to 1 M NaCl (20 mM Tris-HCl, pH 8.0, and 2.5 mM DTT). Following dialysis and a second anion exchange step (ReSource 15Q) Rubisco-containing fractions were applied to a gel filtration column (Superdex 200 Increase 10/300, GE Healthcare) equilibrated with 20 mM Tris-HCl (pH 8.0), 50 mM NaCl, 1 mM DTT, and 5% v/v glycerol, before pure Rubisco fractions were concentrated and stored.

OsRubisco was purified from rice leaves using a combination of ammonium sulfate precipitation, ion-exchange chromatography, and gel filtration[45]. RsRubisco was purified from *E. coli* BL21(DE3) harboring pET30b*RscbbLS* using a sequence of two anion exchange columns (Source30Q and MonoQ) followed by gel filtration[34]. The same protocol was followed to purify Af2Rubisco, Af1Rubisco[33], SeRubisco, the SeRubisco octamer core (SeL8), and SeRubisco-GFP from *E. coli* BL21(DE3) harboring the appropriate plasmid (Supplementary Table 1).

SeRbcL harbored the F345I substitution, which dramatically enhances functional expression in *E. coli*[46]. AfM Rubisco and the Af1Rubisco octamer core was produced as N-terminal His6-Ubiquitin fusion protein and purified using immobilized metal affinity chromatography. The His6-Ubiquitin moiety was cleaved[22] and the protein applied to an anion exchange column followed by size-exclusion chromatography[33].

Rubisco small subunits (CrRbcS and OsRbcS) were produced in *E. coli* BL21 (DE3) harboring pET22b*CrRbcS* and pET22b*OsRbcS* respectively and purified from inclusion bodies[47]. Cells were lysed in buffer B (40 mM Tris-HCl, pH 8.0, 0.25 M sucrose, and 1% v/v Triton X-100) containing 10 mM EDTA and Pierce™ Protease Inhibitor Tablets. The protein pellet was washed using buffer B containing 2 M urea and 40 mM EDTA. The wash was repeated using 40 mM Tris-HCl (pH 8.0), 0.25 M sucrose, and 10 mM EDTA. The washed pellet was dissolved in 40 mM Tris-HCl (pH 7.5), 6 M guanidine-HCl, 1 mM EDTA, and 5 mM DTT. The denatured small subunits were refolded by dialyzing against 50 mM Tris-HCl (pH 8), 50 mM NaCl, and 10 mM MgCl2.

The chimeric Rubisco complexes SeL8CrS8 and SeL8OsS8 were assembled in vitro from the SeL8 large subunit octamer core and refolded small subunits. A unit of 2 µM (active sites) SeL8 were combined with 10 µM of RbcS. The reaction was concentrated 40-fold and excess small subunits were removed using size-exclusion chromatography in buffer A.

**Circular dichroism spectroscopy.** A Chirascan™ Circular Dichroism Spectrometer (Applied Photophysics) using a Quartz cuvette with 0.1 cm pathlength was used to record spectra between 190 and 260 nm in 0.5 nm steps with an averaging time of 2.0 s/step at 25 °C. Protein concentration was 0.15 mg/ml in 20 mM phosphate buffer (pH 8.0). The measured ellipticity, $\theta$, in millidegrees (mdeg) was converted to the mean molar ellipticity per residue, $[\theta]$, in degrees × cm²/dmol by using the equation $[\theta] = \theta \times 100/(c \times l \times N)$, where $c$ is the molar concentration of the sample in mM, $l$ is the pathlength in cm, and $N$ is the number of amino acids.

**Light and fluorescence microscopy.** Reactions (5 µl) were prepared in buffer A unless otherwise indicated. After 3–5 min the solutions were imaged with a Nikon Eclipse Ti Inverted Microscope using the settings for differential interference contrast and epifluorescence microscopy (using fluorescein isothiocyanate filter settings) with a ×100 oil-immersion objective focusing on the coverslip surface. The coverslips used were 22 × 22 mm (Superior Marienfeld, Germany) and fixed in one-well Chamlide CMS chamber for 22 × 22 coverslip (Live Cell Instrument, South Korea). ImageJ was used to pseudocolor all images.

**Droplet sedimentation assay.** EPYC1-Rubisco droplets reconstituted in 10 µl reactions for 10 min (final 20 mM Tris-HCl, pH 8.0, and 50 mM NaCl) were separated at room temperature from the bulk solution by centrifugation for 4–8 min at 21,100 × $g$ (unless indicated otherwise). The imaging software ImageJ/FiJi was used to quantify the intensity of protein bands obtained in the pellet (droplet) and

supernatant (bulk solution) fractions following SDS-PAGE and Coomassie staining. Three gels (technical replicates) were analyzed per quantification.

**Fluorescence recovery after photobleaching**. FRAP was performed using Zen software on a Zeiss LSM710 confocal microscope equipped with a 1.46 numerical aperture (NA) Plan-Apo ×100 oil-immersion lens (Carl Zeiss, Germany). The droplets were scanned five times before photobleaching. Most photobleaching experiments were performed by exposing the region of interest 100 times at 75% intensity of a 488 nm wavelength of an argon laser. The photobleached region was then allowed to recover for 75 cycles, with laser power attenuated to 1.5% intensity. FRAP assays of droplets formed with EPYC1/EPYC1-GFP using $SeL_8$ (Supplementary Fig. 6e) was performed using 100% intensity of the 488 nm laser and allowed to recover for 35 cycles. The area bleached was a 1.25 $\mu m^2$ circle (1.26 $\mu m$ diameter) or varied with droplet size in case the whole droplet was bleached.

Image acquisition and analysis were performed on Zen 2.3 SP1 (black) software from Carl Zeiss. Data were excluded from analysis when droplets fused (causing a spike in fluorescence intensity).

**Estimation of the droplet height to radius ratio**. Super-resolution confocal images were acquired using a LSM710 confocal microscope with a 1.46 NA Plan-Apo ×100 oil immersion lens. Z-stacks of the entire droplets were acquired with 0.5 $\mu m$ intervals and processed using Zen 2.3 SP1 (black) software.

**Rubisco activity assay (continuous)**. Measuring the carboxylase activity of Rubisco was used to quantify functional Rubisco in the supernatant following droplet sedimentation or resuspension of droplets in high salt buffer. The Rubisco used in such experiments was first activated by the cofactors $CO_2$ and $Mg^{2+}$ to form the holoenzyme ECM using a 10 min incubation in activation (ECM) buffer (buffer A, 5 mM $MgCl_2$, and 20 mM $NaHCO_3$) and sedimentation experiments were carried out in ECM buffer. The coupled spectrophotometric assay was used to quantify carboxylase activity[33]. Hundred microliter reactions contained 100 mM Tricine-NaOH (pH 8.0), 5 mM $MgCl_2$, 20 mM $NaHCO_3$, 1 mM DTT/ATP/RuBP, 0.2 mM NADH, 10 mM phosphocreatine, and coupling enzymes[48]. RuBP was synthesized from ribose-5-phosphate[49].

Occasionally, air bubbles were trapped in the light path leading to signal spikes. Such data were excluded from analysis. Experiments utilizing this assay were performed in three or four technical replicates.

**Biochemical Rubisco exchange assay**. Activated Rubisco (ECM—30 $\mu M$) was incubated with 120 $\mu M$ carboxypentitolbisphosphate (50% tight-binding inhibitor CABP, synthesized from RuBP and KCN[32]) to form ECMC, followed by buffer exchange into ECM buffer to remove unbound inhibitor. Droplets were formed using 11 $\mu M$ ECM and 3.7 $\mu M$ EPYC1 in ECM buffer-25 mM NaCl (5.3 $\mu l$ reaction). After 3 min the reaction was supplemented with an equal amount of ECMC in the same buffer (4.1 $\mu l$).

The relative Rubisco activity in the supernatant was determined following droplet sedimentation. For the exchange kinetics experiment (Fig. 3f), time points refer to the period between droplet supplementation and supernatant removal and include centrifugation (21,100 × g, 40 s) and 10 s handling time.

**Two-step discontinuous Rubisco activity assay**. A two-step Rubisco assay was established to measure the activity of demixed Rubisco. In step one, droplet formation of activated Rubisco (ECM) with EPYC1 was triggered 5 min prior to adding RuBP to the demixed enzyme. Upon RuBP addition, the generation of the carboxylation product 3PG was allowed to occur for 15, 30, 45, or 60 s in the reaction mixtures comprising final concentrations of 7.5 $\mu M$ activated Rubisco (ECM active sites), 3.75 $\mu M$ EPYC1, 2.5 mM RuBP, 20 mM Tris-HCl (pH 8.0), 50 mM NaCl, 5 mM $MgCl_2$, and 20 mM $NaHCO_3$. EPYC1-free reactions served as control. The reactions were stopped using 60 $\mu M$ CABP.

3PG was then quantified using the coupled spectrophotometric Rubisco assay. For each reaction, the absorption of 144 $\mu l$ NADH-containing assay mix was measured at 340 nm before and after the addition of 6 $\mu l$ stopped 3PG-containing reactions obtained in step 1. The 3PG concentration was calculated from the resulting reduction in $A_{340}$ (values were corrected for dilution using appropriate controls).

**Native-PAGE gel-shift assay**. A unit of 6 $\mu M$ Rubisco was mixed with 0, 1.5, 3, 6, or 12 $\mu M$ EPYC1Δ2-4 in 20 mM Tris-HCl and 250 mM NaCl, and subjected to 6% native-PAGE.

## Data availability

The datasets generated during and/or analyzed in the current study are available from the corresponding author on reasonable request.

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

## Acknowledgements

We thank O.M.-C lab members for providing diverse Rubisco proteins, Yi-Chin Candace Tsai and Zhijun Guo for constructing some plasmids, and Yansong Miao for insightful discussions. This work was funded by a NTU start-up grant and a Ministry of Education of Singapore Tier 1 grant to O.M.-C. (2016-T1-001-080).

## Author contributions

T.W. and O.M.-C. designed the study. T.W. performed most experiments with help from S.L.H.C. and S.-K.L. S.-K.L. and H.-Y.L. provided expert guidance and assistance with the microscopy work. T.W. and O.M.-C. wrote the manuscript.

## Additional information

**Competing interests:** The authors declare no competing interests.

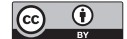

