## [Peer Review File · Nature Communications]

Reviewers' comments:

Reviewer #1 (Remarks to the Author):

Ms nr: NCOMMS-18-15933-T

Title: The phase separation underlying the microalgal Rubisco supercharger

Author(s): Tobias Wunder, Le Hung Cheng, Soak-Kuan Lai, and Hoi-Yeung Li and Oliver Mueller-Cajar

The manuscript by Mueller-Cajar and colleagues describes studies on in vitro formation of phase separation in mixtures of Rubisco and the linker protein, EPYC1.

Eukaryotic microalgae are responsible for a substantial proportion of the global CO₂ fixation. This is achieved by the operation of a carbon-concentration mechanism (CCM) to enhance CO₂ assimilation. Central to the eukaryotic CCM is a non-membrane-bound organelle, the pyrenoid. The presence of the pyrenoid has been known for a long time, but its composition and molecular structure is largely unknown. Recent research has shown that the pyrenoid may behave like a phase-separated liquid compartment, which spatially organises the pyrenoid and dissolves and condenses during the cell cycle, as has been observed before for protein-RNA interactions e.g. in P bodies.

The work by Mueller-Cajar and colleagues presented here investigates the in vitro formation of phase-separated vesicles using recombinant pure preparations of two of the main components of the pyrenoid, Rubisco, and the linker protein EPYC1.

They show that phase separation can be mimicked in vitro using Rubisco and EPYC1 only, that phase separation is dependent on protein- and salt concentration, and that droplet components may exchange rapidly between the droplet and the bulk liquid.

They further show that the extent of phase separation varies in Rubisco from different organisms, and that part of EPYC1 is also able to form droplets in the presence of Rubisco.

These are interesting observations that may be useful for future experiments aiming to engineer carbon concentrating mechanisms into crop plants in order to boost carbon fixation. A number of

questions remain to be answered, for instance, what happens to the in vitro droplet if additional proteins, known to be part of the core of the pyrenoid, are added to the solution? One such protein is Rubisco activase.

The experiments are carefully conducted, described in sufficient detail and well illustrated.

I recommend publication after some minor changes.

Points for improvement:

Title: It is not very descriptive. I suspect that very few readers will understand what is meant by "supercharger" or make the connection to the pyrenoid.

A number of abbreviations have not been specified. It would increase ease of reading if these were explained as they appear.

ECM ECMC may be obvious to readers familiar with Rubisco work, but should be explained in order to promote ease of reading for the general readership.

SeRubisco, CrRubiscoAf1, Af2 AfM, RsRubisco need to be specified as they appear, not just "cyanobacterial Rubisco", or "proteobacterial Rubico", etc.

The conclusion on (top of) page 8, that the L8 core (Rubisco large subunits stripped of the small subunits) interacts directly with EPYC1 is not well founded. It is known that the L8 core is hydrophobic in nature and that it easily falls out of solution. The abnormal clumped droplets observed here may just be an effect of the increased hydrophobicity of the L subunit.

Figure 1 e. The colours (grey-green-black) indicating % RbcL recovery is difficult to discern.

Reviewer #2 (Remarks to the Author):

The article by Wunder, et al., reports that *C. reinhardtii* Rubisco (CrRubisco) and EPYC1, a known component of chloroplasts, undergo liquid-liquid phase separation (LLPS) in vitro at low micromolar concentrations through interactions that are inhibited at moderate NaCl concentrations. The droplets that formed have liquid-like features as shown by their ability to fuse and because their components are mobile as shown by FRAP. The ability of a variety of Rubisco enzymes from different organisms to undergo LLPS with EPYC1 was tested, with some positive for LLPS and others negative. The authors studied the so-called repeat domain of EPYC1, with 4 repeats of a PAS-rich sequence, which is intrinsically disordered based on CD data and sequence analysis. The authors state that LLPS occurs through complex coacervation, in part based on the NaCl concentration dependence of LLPS, and also due to what they say is charge complementarity between negatively charged CrRubisco and positively charged EPYC1. The authors propose that Rubisco and EPYC1 interact through phase separation to form the pyrenoid bodies in which CrRubisco functions. While the authors do demonstrate LLPS by CrRubisco and EPYC1 in vitro, the manuscript falls short with regard to establishing the relevance of this observation to the biological structural state of these two proteins in chloroplasts. Many oppositely charged proteins can undergo complex coacervation in vitro, but this observation alone does not establish biological relevance. Also, the authors do not show that CrRubisco is catalytically active in the phase separated droplets with EPYC1; providing these data would enhance the biological relevance of their article. In the current form, the article presents an interesting observation certainly worthy of further investigation but which falls short in terms of demonstrated biological relevance for publication at Nat. Comm.

Additional comments:

1. While the sequence of EPYC1 is positively charged, as the authors note, there are repeats of amino acids such as Pro, Ser and Ala, as well as a few Glu's, that are likely contributing to LLPS. Most of the basic residues are Arg's, and these may be playing a role in LLPS. The EPYC1 sequence displays acidic residues that alternate with stretches of basic residues; these features may promote homotypic LLPS under crowded (e.g., PEG-8K) conditions; have the authors explored this possibility? The CIDER web site of Rohit Pappu can be used to analyze the charge patterning within proteins and may be helpful to the authors in these studies.
2. Is it known whether knock-down of EPYC1 affects the structure and function of pyrenoid bodies in chloroplasts? If not, the authors may consider trying to develop such data. If it is possible to express exogenous EPYC1 in the knock-down setting, perhaps phase-separation-deficient mutants could be expressed to test the LLPS hypothesis. Mutations could be to Arg residues, as well as acidic and Pro residues (separately) within the repeat domain, to test their roles in LLPS. Papers from the Pappu and Chilkoti labs might provide guidance.

Reviewer #3 (Remarks to the Author):

The pyrenoid, the eukaryotic algae CO₂ fixing organelle, is responsible for nearly one-third of global CO₂ fixation yet until recently our understanding of pyrenoid function was very limited. In addition, the ability to engineer a pyrenoid based CO₂ concentrating mechanism into C₃ crop plants could lead to significant yield gains. In this study, Mueller-Cajar and colleagues show for the first time the in vitro liquid-liquid phase separation (LLPS) of two core and abundant pyrenoid components, EPYC1 and Rubisco. They show that EPYC1 and Rubisco are the only necessary components needed for phase separation, that it is dependent on ionic strength and that EPYC1 repeat number is important. By looking at LLPS using different Rubisco forms with different small subunits they give interesting insights in the binding compatibilities of EPYC1. This study provides a very timely and significant advance in our knowledge of the pyrenoid and provides a strong experimental platform to test in vitro pyrenoid assemblies and component compatibilities prior to plant engineering. I believe it will have a considerable impact in the fields of photosynthesis, biological CO₂ fixation and cellular liquid-liquid phase separation. Furthermore, experiments are well designed and well executed, the manuscript is well written, and the figures crisply presented.

My comments are generally minor, with my only concern related to the interpretation of the data related to the EPYC1 binding site being on the Rubisco large subunit. Below I suggest some potential additional experimental comparisons and further analysis to strengthen the authors data.

Minor comments:

Introduction lines 22-24, lines 157-161 and lines 189-193: The authors propose that the large subunit (LSU) is the primary EPYC1 binding site. I agree that some of their data presented in Figure 3 supports this, however my primary concern is that potentially in the absence of the availability of EPYC1s normal Rubisco small subunit binding site, EPYC1 binds to a different region of the large subunit (that may not be exposed in correct L8S8 assembly) resulting in aberrant complex formation giving protein aggregations that lack the shape and dynamic properties of the CrL8S8-EPYC1 complex (as seen in Fig 3D, Ext data Fig 4 and 5). There are also three bits of data presented in 3A and C that could be interpreted as supporting this potentially aberrant binding:

1) There is considerably reduced LLPS/aggregation in SeL8OsS8 vs SeL8CrS8. If the LSU contained the EPYC1 binding region you would expect equal LLPS.

2) Removal of an “incompatible” small subunit results in uncontrolled aggregation, as shown in SeL8OsS8 vs SeL8.

3) Removal of RbcS from Af1 results in Rubisco aggregation.

Related to this, it would be interesting to see the characteristics of the phase separated non-spherical droplets of SeL8-EPYC1 and SeL8OsS8-EPYC1. Do they show the same internal and bulk mixing properties as CrL8S8-EPYC1 or SeL8S8-EPYC1? If the large subunit is the true binding site we might expect similar dynamics, however if a binding site is in the small subunit or at a large-small subunit interface different properties related to the dynamics maybe observed.

Additional comparisons to further help elucidate the binding site would be to look at the LLPS of EPYC1 with the Chlamydomonas LSU-higher plant SSU hybrids used by Meyer et al 2012 (PNAS). Or to see if Chlamydomonas SSU on its own is enough to induce phase separation.

Line 185 (and related to above): “Instead both EPYC1, which is positively charged, and negatively charged Rubisco are essential for the process to take place, which is salt sensitive”. It would be good to have some data to support this statement. An in silico structural comparison of the Rubisco forms used (i.e. Cr, Os, Se, SeL8 w/o S8, Af1, Af1 w/o S8) in particular looking at Rubisco (and EPYC1) surface charge could give an interesting insight into likely EPYC1 binding sites and potentially identify binding sites that become available in the absence of the small subunit.

Line 198: Maybe further expand this point related to the magic number effect. The magic number effect indicates that 4 repeats would lead to a dissolution at a 1:1 EPYC1 to Rubisco binding site ratio, whereas 5 repeats or 3 repeats would result in aggregation at similar ratios. If EPYC1 has an additional Rubisco binding site in the C-terminus – as indicated by the presented data - it would mean that the number of binding sites tested via your truncation experiments would be 5,3 and 2. I think you are in a very strong position to test the magic number effect by creating an EPYC1 with just a single repeat deleted (4 binding sites) and analysing the propensity for LLPS by varying Rubisco EPYC1 ratios as done in Fig 4 and Ext Data Fig 6.

Further comments:

Line 51: use CCMs (acronym introduced earlier in text)

Line 58: LLPS acronym already given in introduction

Ext Data Fig 1: Need to expand abbreviations of different Rubisco's in legend.

Line 124: Unclear use of acronyms and parenthesis.

Figure 2g: Labelling below panel is not clear. It could be removed and explained in the legend or modified to show aspect of time.

Figure 3C: What Rubisco concentration was used. Not mentioned in figure or figure legend. I assume 15 μM ?

Line 183-185: "EPYC1 was previously predicted to form the pyrenoid scaffold, but unexpectedly it does not phase separate and form droplets in isolation". Please add a citation. The original work done by Mackinder et al 2016 (PNAS), proposed two possible functions 1) A scaffold and 2) A link between separate Rubisco enzymes (as shown to be the case by your work).

Response to reviewers

We thank all the reviewers for their insightful comments.

Reviewer #1 (Remarks to the Author):

Points for improvement:

Title: It is not very descriptive. I suspect that very few readers will understand what is meant by "supercharger" or make the connection to the pyrenoid.

We have changed the title to now include the pyrenoid.

A number of abbreviations have not been specified. It would increase ease of reading if these were explained as they appear.

ECM ECMC may be obvious to readers familiar with Rubisco work, but should be explained in order to promote ease of reading for the general readership.

We have now included a clarifying sentence prior to introducing these concepts as follows:

“To become catalytically functional, Rubisco active sites need to bind CO₂ and Mg²⁺ cofactors to form the holoenzyme termed ECM. ECM has an extraordinary high affinity for the carboxylation transition state analogue carboxyarabinitol bisphosphate³⁰ (CABP), permitting dead-end inhibited complexes (ECMC) to be formed.”

SeRubisco, CrRubiscoAf1, Af2 AfM, RsRubisco need to be specified as they appear, not just "cyanobacterial Rubisco", or "proteobacterial Rubico", etc.

These have now been explained in more detail in the text.

The conclusion on (top of) page 8, that the L8 core (Rubisco large subunits stripped of the small subunits) interacts directly with EPYC1 is not well founded. It is known that the L8 core is hydrophobic in nature and that it easily falls out of solution. The abnormal clumped droplets observed here may just be an effect of the increased hydrophobicity of the L subunit.

Concerning this point and that of reviewer 3 we have investigated this effect more closely (Please see comments for Reviewer 3 for more detail). In brief we come to the conclusion that the SeL₈ core does indeed interact (and phase separate) with EPYC1, however we now acknowledge the newly exposed surfaces likely contribute to this interaction.

Figure 1 e. The colours (grey-green-black) indicating % RbcL recovery is difficult to discern. We have changed the colour scheme.

Reviewer #2 (Remarks to the Author):

The article by Wunder, et al., reports that *C. reinhardtii* Rubisco (CrRubisco) and EPYC1, a known component of chloroplasts, undergo liquid-liquid phase separation (LLPS) in vitro at low micromolar concentrations through interactions that are inhibited at moderate NaCl concentrations. The droplets that formed have liquid-like features as shown by their ability to fuse and because their components are mobile as shown by FRAP. The ability of a variety of Rubisco enzymes from different organisms to undergo LLPS with EPYC1 was tested, with some positive for LLPS and others negative. The authors studied the so-called repeat domain of EPYC1, with 4 repeats of a PAS-rich sequence, which is intrinsically disordered based on CD data and sequence analysis. The authors state that LLPS occurs through complex coacervation, in part based on the NaCl concentration dependence of LLPS, and also due to what they say is charge complementarity between negatively charged CrRubisco and positively charged EPYC1. The authors propose that Rubisco and EPYC1 interact through phase separation to form the pyrenoid bodies in which CrRubisco functions. While the authors do demonstrate LLPS by CrRubisco and EPYC1 in vitro, the manuscript falls short with regard to establishing the relevance of this observation to the biological structural state of these two proteins in chloroplasts. Many oppositely charged proteins can undergo complex coacervation in vitro, but this observation alone does not establish biological relevance. Also, the authors do not show that CrRubisco is catalytically active in the phase separated droplets with EPYC1; providing these data would enhance the biological relevance of their article.

We have now included an experiment (Fig. 2b) comparing the time course of production of the carboxylation product 3-phosphoglycerate in reactions containing Rubisco and EPYC1 (demixed) and Rubisco alone. The result demonstrates equivalent enzymatic activity for the phase-separated enzyme, and suggests that the process of interaction with EPYC1 and inclusion in phase separated droplets does not greatly influence catalytic activity. In the discussion we point out that this is as expected, since the CO₂ concentrating mechanism relies on increasing CO₂ concentration at the Rubisco active site, not on enhancing Rubisco kinetics directly.

In the current form, the article present an interesting observation certainly worthy of further investigation but which fall short in terms of demonstrated biological relevance for publication at Nat. Comm.

Additional comments:

1. While the sequence of EPYC1 is positively charged, as the authors note, there are repeats of amino acids such as Pro, Ser and Ala, as well as a few Glu's, that are likely contributing to LLPS. Most of the basic residues are Arg's, and these may be playing a role in LLPS. The EPYC1 sequence displays acidic residues that alternate with stretches of basic residues; these features may promote homotypic LLPS

under crowded (e.g., PEG-8K) conditions; have the authors explored this possibility? The CIDER web site of Rohit Pappu can be used to analyze the charge patterning within proteins and may be helpful to the authors in these studies.

The original manuscript does explore the possibility of homotypic LLPS of EPYC1 at quite high concentrations (150 μM – Extended data Fig. 2h). We have now exacerbated this condition, and find that indeed at 200 μM EPYC1, it is possible to observe some material forming droplets and pelleting in the presence of 10% w/v PEG. However this condition is far removed from the 7.5 μM EPYC1, 30 μM Rubisco active site concentrations utilized to achieve complete phase separation when both components are present. We have now included this data (Supplementary Fig. 2h and i) and made the appropriate modification to the text.

We have now used the CIDER website to address the question of reviewer 3, regarding the distribution of positive charges in EPYC1 (Supplementary Fig. 9b).

2. Is it known whether knock-down of EPYC1 affects the structure and function of pyrenoid bodies in chloroplasts? If not, the authors may consider trying to develop such data. If it is possible to express exogenous EPYC1 in the knock-down setting, perhaps phase-separation-deficient mutants could be expressed to the LLPS hypothesis. Mutations could be to Arg residues, as well as acidic and Pro residues (separately) within the repeat domain, to test their roles in LLPS. Papers from the Pappu and Chilkoti labs might provide guidance.

Yes, the physiological effect of EPYC1 reduction is well described, and we now summarize this work in the introduction by including the following key results reported by MacKinder et al. (2016). “Importantly the *epyc1* mutant, which contains severely reduced levels of this protein, requires high CO_2 for photoautotrophic growth and is unable to effectively concentrate CO_2 . The pyrenoid of *epyc1* is of decreased size, matrix density is reduced and the majority of Rubisco is localized to the chloroplast stroma”

The proposed experiments (regarding phase separation deficient mutants in vivo) are clearly of great interest to us, we would indeed aim to develop such data in future collaborative work. The suggested dissection of EPYC1 sequence motifs is also planned, however, we consider it outside of the scope of this current first manuscript.

Reviewer #3 (Remarks to the Author):

The pyrenoid, the eukaryotic algae CO_2 fixing organelle, is responsible for nearly one-third of global CO_2 fixation yet until recently our understanding of pyrenoid function was very limited. In addition, the ability to engineer a pyrenoid based CO_2 concentrating mechanism into C_3 crop plants could lead to significant yield gains. In this study, Mueller-Cajar and colleagues show for the first time the in vitro liquid-liquid phase separation (LLPS) of two core and abundant pyrenoid components, EPYC1 and Rubisco. They show that EPYC1 and Rubisco are the only necessary components needed for phase separation, that it is dependent on ionic strength and that EPYC1 repeat number is important. By looking at LLPS using different Rubisco forms with different small subunits they give interesting insights in the

binding compatibilities of EPYC1. This study provides a very timely and significant advance in our knowledge of the pyrenoid and provides a strong experimental platform to test in vitro pyrenoid assemblies and component compatibilities prior to plant engineering. I believe it will have a considerable impact in the fields of photosynthesis, biological CO₂ fixation and cellular liquid-liquid phase separation. Furthermore, experiments are well designed and well executed, the manuscript is well written, and the figures crisply presented.

My comments are generally minor, with my only concern related to the interpretation of the data related to the EPYC1 binding site being on the Rubisco large subunit. Below I suggest some potential additional experimental comparisons and further analysis to strengthen the authors data.

Minor comments:

Introduction lines 22-24, lines 157-161 and lines 189-193: The authors propose that the large subunit (LSU) is the primary EPYC1 binding site. I agree that some of their data presented in Figure 3 supports this, however my primary concern is that potentially in the absence of the availability of EPYC1s normal Rubisco small subunit binding site, EPYC1 binds to a different region of the large subunit (that may not be exposed in correct L8S8 assembly) resulting in aberrant complex formation giving protein aggregations that lack the shape and dynamic properties of the CrL8S8-EPYC1 complex (as seen in Fig 3D, Ext data Fig 4 and 5). There are also three bits of data presented in 3A and C that could be interpreted as supporting this potentially aberrant binding:

1) There is considerably reduced LLPS/aggregation in SeL8OsS8 vs SeL8CrS8. If the LSU contained the EPYC1 binding region you would expect equal LLPS.

2) Removal of an “incompatible” small subunit results in uncontrolled aggregation, as shown in SeL8OsS8 vs SeL8.

3) Removal of RbcS from Af1 results in Rubisco aggregation.

Related to this, it would be interesting to see the characteristics of the phase separated non-spherical droplets of SeL8-EPYC1 and SeL8OsS8-EPYC1. Do they show the same internal and bulk mixing properties as CrL8S8-EPYC1 or SeL8S8-EPYC1? If the large subunit is the true binding site we might expect similar dynamics, however if a binding site is in the small subunit or at a large-small subunit interface different properties related to the dynamics maybe observed.

Additional comparisons to further help elucidate the binding site would be to look at the LLPS of EPYC1 with the Chlamydomonas LSU-higher plant SSU hybrids used by Meyer et al 2012 (PNAS). Or to see if Chlamydomonas SSU on its own is enough to induce phase separation.

Both reviewer 1 and 3 point out that our conclusion that the large subunit of Rubisco provides the EPYC1 binding site may be premature, in particular because the L₈-core now exposes a new surface that may

interact with EPYC1 non-specifically. We have therefore expanded our analysis of this effect and come to the conclusion that the reviewers' interpretation is likely to be correct. We have revised the manuscript as detailed below.

Specifically we now include the following data prompted by the following suggestions:

Additional comparisons to further help elucidate the binding site would be to look at the LLPS of EPYC1 with the *Chlamydomonas* LSU-higher plant SSU hybrids used by Meyer et al 2012 (PNAS). Or to see if *Chlamydomonas* SSU on its own is enough to induce phase separation.

Chlamydomonas SSU alone did not induce phase separation, however this could be due to a lack of multivalency. We now include this data in the manuscript (Supplementary Fig. 5c,d).

Related to this, it would be interesting to see the characteristics of the phase separated non-spherical droplets of SeL8-EPYC1 and SeL8OsS8-EPYC1. Do they show the same internal and bulk mixing properties as CrL8S8-EPYC1 or SeL8S8-EPYC1? If the large subunit is the true binding site we might expect similar dynamics, however if a binding site is in the small subunit or at a large-small subunit interface different properties related to the dynamics maybe observed.

This experiment has now been performed, and suggests that the mixing dynamics are similar for the aberrant droplets (Supplementary Fig. 6d,e).

To add an additional dimension to the interaction between the different Rubiscos and EPYC1, we decided to introduce a protein-protein interaction assay (Fig. 4f). Here we are measuring the relative mobility of Rubisco using Native-PAGE after being exposed to the minimal EPYC1 fragment EPYC1 Δ 2-4. EPYC1 Δ 2-4 does not enter this gel system in isolation, as it is positively charged, but it retards the migration of phase separating Rubisco (CrLS) but not OsLS. This experiment suggested that both SeL8 and AfL8 interacted more strongly with the EPYC1 fragment than the holoenzymes. We conclude that the enzymes lacking the small subunits are more prone to bind EPYC1, indicating that the reviewer is likely correct.

We have adjusted the manuscript in the relevant places to reflect this change in interpretation.

The key section affected in the Results section is the following:

“The observed demixing of the SeL₈ core could be driven either by specific interactions between EPYC1 and the Rubisco large subunit or non-specific interactions with the surface covered by the small subunit in the holoenzyme. Adding an EPYC1 fragment (EPYC1 Δ 2-4, containing the first repeat and the C-terminus, Supplementary Table 1) to Rubisco followed by Native-PAGE results in a smear and reduced mobility for CrLS, but not OsLS, and thus provides a protein-binding assay (Fig. 4f, Supplementary Fig. 7). Note that EPYC1 Δ 2-4 alone does not enter the gel due to its positive charge. This gel-shift assay indicated that both AfL₈ and SeL₈ cores interacted more strongly with EPYC1 Δ 2-4 than the respective L₈S₈ holoenzymes (Fig. 4f, Supplementary Fig 7). This comparison implies that the newly exposed large subunit surface is likely to contribute to the interaction in both cases.”

Line 185 (and related to above): “Instead both EPYC1, which is positively charged, and negatively charged Rubisco are essential for the process to take place, which is salt sensitive”. It would be good to have some data to support this statement. An in silico structural comparison of the Rubisco forms used (i.e. Cr, Os, Se, SeL8 w/o S8, Af1, Af1 w/o S8) in particular looking at Rubisco (and EPYC1) surface charge could give an interesting insight into likely EPYC1 binding sites and potentially identify binding sites that become available in the absence of the small subunit.

To provide more context to this statement, we now visualize the surface charge of experimentally available Rubisco structures (Cr, Os, Se, SeL8) (Supplementary Fig. 9a). We also analyze the charge distribution on EPYC1 using the CIDER webtool proposed by Reviewer 2. The analysis is summarized in the discussion, since it is expected to provide directions for future work.

Line 198: Maybe further expand this point related to the magic number effect. The magic number effect indicates that 4 repeats would lead to a dissolution at a 1:1 EPYC1 to Rubisco binding site ratio, whereas 5 repeats or 3 repeats would result in aggregation at similar ratios. If EPYC1 has an additional Rubisco binding site in the C-terminus – as indicated by the presented data - it would mean that the number of binding sites tested via your truncation experiments would be 5,3 and 2. I think you are in a very strong position to test the magic number effect by creating an EPYC1 with just a single repeat deleted (4 binding sites) and analysing the propensity for LLPS by varying Rubisco EPYC1 ratios as done in Fig 4 and Ext Data Fig 6.

To address this comment we have generated two additional EPYC1 variants containing 3 and 5 repeats and include them in our analysis. In summary we do not find support for the magic number theory proposed earlier, but find that 3-5 EPYC1 repeats demix Rubisco equally well. An increased number of repeats does correlate with an increased height to radius ratio for droplets, as well as slightly reduced EPYC1-GFP mobility.

Further comments:

Line 51: use CCMs (acronym introduced earlier in text)

The change has been made

Line 58: LLPS acronym already given in introduction

The change has been made

Ext Data Fig 1: Need to expand abbreviations of different Rubisco's in legend.

The change has been made

Line 124: Unclear use of acronyms and parenthesis.

This issue has been addressed – ECM and ECMC is now introduced and explained earlier as part of the experiments addressing reviewer #2.

Figure 2g: Labelling below panel is not clear. It could be removed and explained in the legend or modified to show aspect of time.

We have addressed this issue as suggested, and modified both the Figure and the legend (now Fig. 3e)

Figure 3C: What Rubisco concentration was used. Not mentioned in figure or figure legend. I assume 15 μM ?

We now indicate the concentrations.

Line 183-185: “EPYC1 was previously predicted to form the pyrenoid scaffold, but unexpectedly it does not phase separate and form droplets in isolation”. Please add a citation. The original work done by Mackinder et al 2016 (PNAS), proposed two possible functions 1) A scaffold and 2) A link between separate Rubisco enzymes (as shown to be the case by your work).

We have expanded our explanation here as follows:

“EPYC1 was predicted to either assemble into a pyrenoid scaffold to which Rubisco binds, or form a codependent network with the enzyme¹⁴. In contrast to the scaffold hypothesis we find EPYC1 does not phase separate and form droplets in isolation. Instead both EPYC1 and Rubisco are essential for the process to take place, supporting the co-dependent network model.”

In addition to these points we have made appropriate changes to the methods to comply with reporting requirements. We have included two more relevant key references – Zhan et al. Plos One 2018 and Long et al. 2018.

REVIEWERS' COMMENTS:

Reviewer #2 (Remarks to the Author):

The revised manuscript from Wunder, et al., is improved over the original in many respects and, importantly, clarifies the mechanism through which EPYC1 mediates LLPS with Rubisco through addition of new data (in Fig. 5). Also, the new functional data under LLPS conditions enhance the significance of the findings (Fig. 2). The manuscript significantly advances understanding of how LLPS concentrates Rubisco in liquid-like pyrenoid structures. This reviewer requests one additional, minor change to the revised manuscript. On page 12 in the Discussion, the authors state that EPYC1 does not undergo LLPS, as follows.

Lines 244-245. "In contrast to the scaffold hypothesis we find EPYC1 it does not phase separate and form droplets in isolation."

However, the authors' in vitro data shows that it does, although at concentrations that are not physiologically relevant. The reviewer suggests revision, as follows.

"In contrast to the scaffold hypothesis, we find that EPYC1, while able to homotypically phase separate in vitro at supra-physiological concentrations, does not do so at concentrations found in cells."

It is important to be accurate in reporting these data. While EPYC1 does not readily undergo homotypic LLPS, these interactions likely contribute in a minor way to the LLPS that is observed under physiological conditions (when heterotypic LLPS is the crowding mechanism).

With this minor revision, the manuscript will be suitable for publication.

Reviewer #3 (Remarks to the Author):

Mueller-Cajar and colleagues have thoroughly addressed all of the comments raised by the reviewers. I believe they have considerably further strengthened the manuscript. In particular related to the EPYC1 large subunit interactions and testing additional EPYC1 repeat numbers.

This is a novel and exciting piece of work backed by detailed and well executed experiments.

Response to Reviewers

We are happy to see that our revisions have been assessed favourably.

Reviewer 2:

On page 12 in the Discussion, the authors state that EPYC1 does not undergo LLPS, as follows.

Lines 244-245. “In contrast to the scaffold hypothesis we find EPYC1 it does not phase separate and form droplets in isolation.”

However, the authors’ in vitro data shows that it does, although at concentrations that are not physiologically relevant. The reviewer suggests revision, as follows.

“In contrast to the scaffold hypothesis, we find that EPYC1, while able to homotypically phase separate in vitro at supra-physiological concentrations, does not do so at concentrations found in cells.”

It is important to be accurate in reporting these data. While EPYC1 does not readily undergo homotypic LLPS, these interactions likely contribute in a minor way to the LLPS that is observed under physiological conditions (when heterotypic LLPS is the crowding mechanism).

To address this point we have modified the discussion as follows to more accurately reflect the results:

“In contrast to the scaffold hypothesis we find EPYC1 is relatively soluble and only homotypically phase separates at concentrations of 100 μ M or more in the presence of a crowding agent. To achieve efficient phase separation at lower concentrations both EPYC1 and Rubisco are essential, supporting the co-dependent network model.”

We have chosen not to mention the physiological concentration of EPYC1 here, since no *in vivo* quantification has been reported.